# Design of Obstacle Avoidance for Autonomous Vehicle Using Deep Q-Network and CARLA Simulator

**Wasinee Terapaptommakol** [1,2], **Danai Phaoharuhansa** [1,*], **Pramote Koowattanasuchat** [2,*] and **Jartuwat Rajruangrabin** [2,*]

1   Department of Mechanical Engineering, King Mongkut's University of Technology Thonburi, Bangkok 10140, Thailand
2   Rail and Modern Transports Research Center, National Science and Technology Development Agency, Pathum Thani 12120, Thailand
*   Correspondence: danai.pha@kmutt.ac.th (D.P.); pramote.koo@nstda.or.th (P.K.); jartuwat.raj@nstda.or.th (J.R.)

**Abstract:** In this paper, we propose a deep Q-network (DQN) method to develop an autonomous vehicle control system to achieve trajectory design and collision avoidance with regard to obstacles on the road in a virtual environment. The intention of this work is to simulate a case scenario and train the DQN algorithm in a virtual environment before testing it in a real scenario in order to ensure safety while reducing costs. The CARLA simulator is used to emulate the motion of the autonomous vehicle in a virtual environment, including an obstacle vehicle parked on the road while the autonomous vehicle drives on the road. The target position, real-time position, velocity, and LiDAR point cloud information are taken as inputs, while action settings such as acceleration, braking, and steering are taken as outputs. The actions are sent to the torque control in the transmission system of the vehicle. A reward function is created using continuous equations designed, especially for this case, in order to imitate human driving behaviors. The results demonstrate that the proposed method can be used to navigate to the destination without collision with the obstacle, through the use of braking and dodging methods. Furthermore, according to the trend of DQN behavior, a better result can be obtained with an increased number of training episodes. This method is a non-global path planning method successfully implemented on a virtual environment platform, which is an advantage of this method over other autonomous vehicle designs, allowing for simulation testing and application with further experiments in future work.

**Keywords:** obstacle avoidance; autonomous vehicle; deep Q-network; CARLA simulator

## 1. Introduction

With the growth of the automotive industry, road traffic injuries in Thailand are increasing every year, with over 13,000 deaths and a million injuries each year [1]. Most road traffic accidents are caused by human error. Autonomous vehicles are already being sold in the public market, which can assist drivers through technologies such as adaptive cruise control, automatic parking control, and lane-keep assist systems.

Thailand has one of the highest rates of injuries and fatalities due to car accidents worldwide. The main reason for this is that drivers do not avoid obstacles in time, and about 20,000 people die in road accidents each year [1,2]. The research paper [3] presented statistics regarding the collision of a car with an obstacle in front, indicating that such a scenario will cause a risk of injury in up to almost 100% of cases, and leads to a high risk of death. Due to human error leading to injury and losses, autonomous vehicles were developed to address this problem.

To improve autonomous control systems, one of the important issues is the consideration of the safety of the driver and passengers. Conventional control systems are based on mathematical models [4], but they only control the vehicle in a limited range of situations.

Therefore, machine learning algorithms have been applied in autonomous systems [5,6] in order to better control vehicles in varied situations.

For on-site experiments, tests need to be conducted on real traffic roads, which leads to high costs for the preparation of the experiment, such as environmental control and the measurement system. Therefore, simulation software has been extensively developed to simulate the motion of vehicles in virtual environments before conducting experiments in actual conditions.

Thus, the main motivation of this study is to reduce traffic fatality rates and accidents on the road by testing the proposed method in a virtual environment. For this purpose, we design and validate the hyperparameters and reward function affecting the behavior of the DQN model to make decisions in the context of autonomous control. Therefore, this study has value as a guideline for simulating and validating autonomous vehicle motion, and the results will be applied to other methods in the future, including on-site experiments.

## 2. Related Work

In this section, we briefly review reference works in the literature related to this project, such as those focused on object detection using LiDAR point clouds, simulation programs, and a fusion of them. Over the past few years, autonomous vehicles have been tested using the Deep Q learning technique [7]. The advantage of this technique is that it can learn through training of the algorithm using huge sample spaces to provide solutions to complex problems. In the research paper [8], a camera was used to detect the distance from the vehicle to obstacles; however, this led to problems related to information uncertainty, and the error is higher than that when using information from a LiDAR sensor.

Referring to [9,10], both papers have focused on autonomous vehicles and provided algorithms based on a Deep Q-Network (DQN) for the control layer, which was simulated in the CARLA simulator to avoid any risk during testing. The results indicated that the use of image data had an effect on the calculation process in the DQN algorithm, as it has a low frequency of information collection. For this purpose, LiDAR point clouds are suitable for the detection of objects more than images. Therefore, we found it reasonable to apply LiDAR in this project. The LiDAR point cloud boundary information was assessed in the scope of the detection space, demonstrating that the distance between the vehicle and obstacles is sufficient to avoid collisions.

The DQN algorithm uses this information to make decisions, based on the reward function [11]. As mentioned above, the purpose of this project is to use LiDAR sensor data to detect objects, while the DQN algorithm is applied as an avoidance control system, in order to avoid obstacles. Both research papers above focused on adjusting the algorithm to succeed in the same task using different models. Therefore, in this paper, we focus on adjusting the reward function of the DQN algorithm to succeed in the mission while ensuring smooth motion.

The Gazebo and CARLA simulators are open-source software, which can be used to simulate autonomous vehicle motion with 3D visualization. In [12], Gazebo was used to estimate the performance prior to real track tests, due to limitations in graphics quality and physics engine. The CARLA software has been utilized in several papers [9,10,13], and it may be better than Gazebo as it has been specifically prepared for testing autonomous vehicles, including physics engines and sensors that can adjust for any noise. It has been developed from the ground up to support the development, training, and validation of autonomous driving systems by simulating the response of vehicles in virtual situations. The advantages are that the program can obtain a certain physics engine and real-world map, including many assets to create the experimental conditions, such as any type of vehicle, traffic attributes, onboard sensors, and environment for the autonomous system. It supports the flexible specification of sensor suites, virtual environmental conditions, full control of static and dynamic actors, map generation, and much more, through the use of python code for control.

### 3. Autonomous Vehicle

Autonomous vehicle technology has been evolving rapidly with the continuous improvement of artificial intelligence technology. Its performance on highways with the exact route path has been described [14]. Autonomous driving systems consist of two main parts [15], as shown in Figure 1: hardware and software [16]. The hardware part includes the mechanical components, such as sensors, Vehicle-to-Vehicle (V2V) hardware, and actuators, while sensors are used to observe the environment. V2V hardware is set up on each vehicle for communication and the sharing of data between the vehicles, while actuators are used to power the sub-systems of the vehicle, such as the power train, steering wheel, and brakes. The software modules include perception, path planning, and control systems [15]. The perception module collects data from the sensors and builds a 3D mapping of the environment around the vehicle, allowing the car to understand the environment, as depicted in Figure 2. Then, path planning design can be conducted, based on user commands. Finally, the control system sends action orders to the vehicle hardware after observing the actual environment.

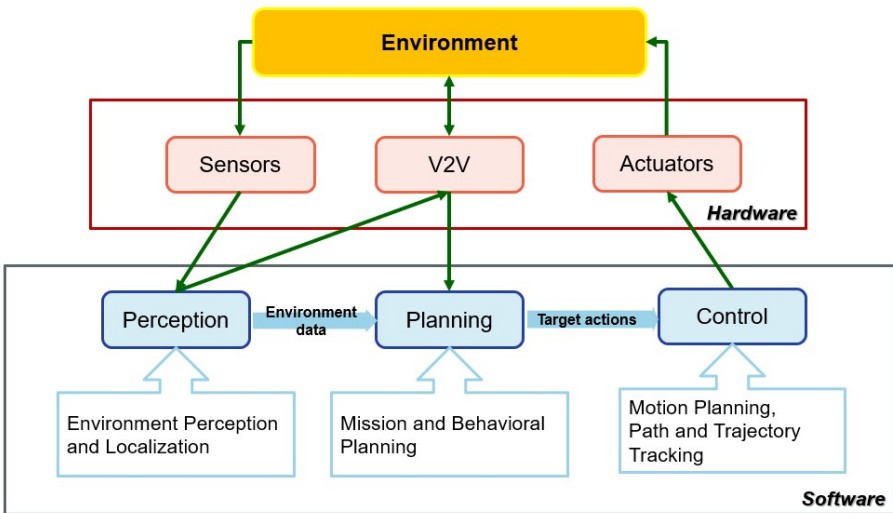

**Figure 1.** Architecture of autonomous driving system.

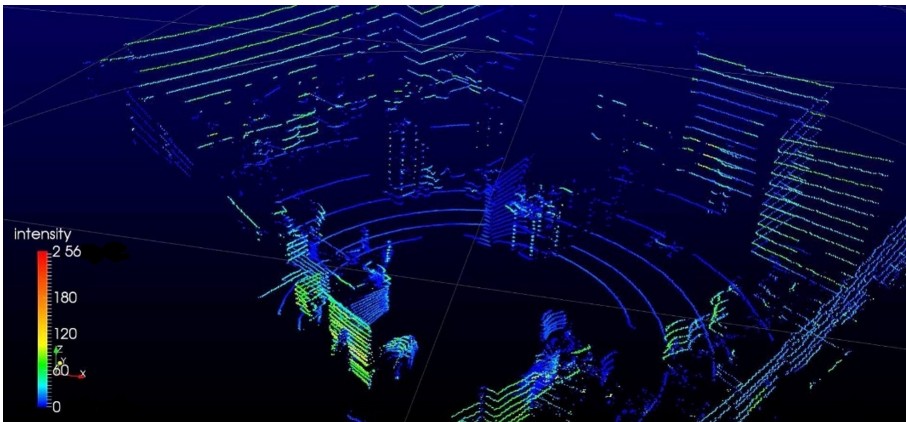

**Figure 2.** Example of point cloud output from a LiDAR.

The onboard sensors consist of several types of sensors, such as cameras, radar, and LiDAR sensors. Camera sensors in automated driving systems have two concern points: The sampling rate and response to visibility conditions.

Radar technology uses high-frequency electromagnetic waves to measure the distance to objects based on the round-trip time principle. It is independent of light and weather conditions and can measure up to 250 m under very adverse conditions.

Light Detection and Ranging (LiDAR) is an active-ranging technology that calculates the distance to objects by measuring the round-trip time of a laser light pulse [17]. Laser beams have a low divergence for reducing power decay, and the measured distance ranges up to 200 m under direct sunlight. A 3D point cloud is shown in Figure 2, which represents a 3D visualization of the environment generated according to information obtained from the LiDAR [18].

## 4. CARLA Simulator

CARLA (Car Learning to Act) [19,20] is an open-source software for urban driving environments, which was developed in order to simulate vehicles and vehicle control systems, in order to support the training, prototyping, and validation of autonomous driving models [21]. The simulator includes visualization relative to the vision of sensors and third-person vision, as shown in Figure 3a,b, respectively. The process can be separated to apply autonomous vehicle control simulation into three components, as shown in Figure 4: Virtual environment, agent, and control system.

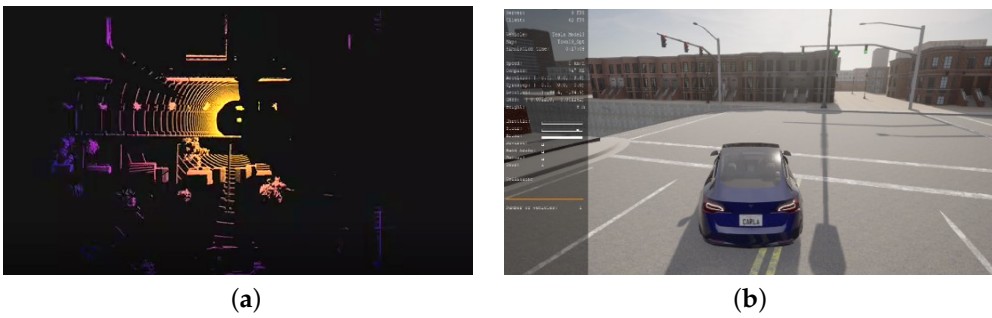

| (**a**) | (**b**) |

**Figure 3.** CARLA simulator presents by LiDAR information and 3D visualization. (**a**) The LIDAR information. (**b**) Third person visualization.

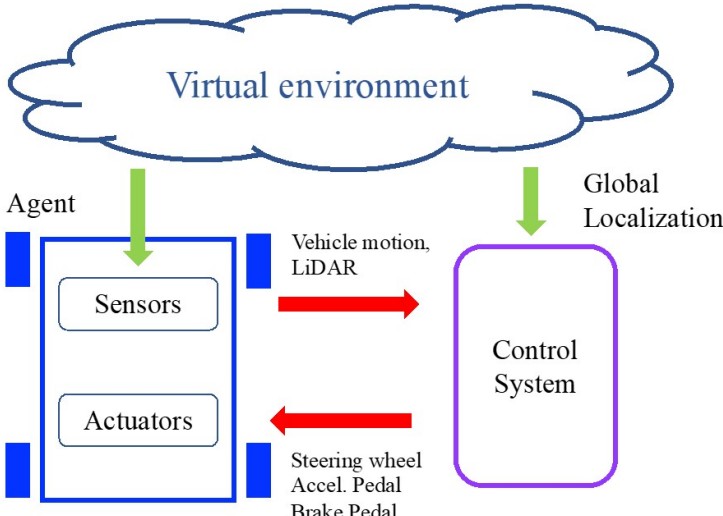

**Figure 4.** CARLA architecture for autonomous vehicle simulation.

### 4.1. Virtual Environment

The virtual environment is presented as 3D models, which consist of static and dynamic objects. Static objects include the road, buildings, traffic signs, and infrastructure, while dynamic objects denote moving objects in the environment, such as pedestrians and other vehicles. All models are designed to reconcile visual quality and rendering speed. All environment models can be defined in terms of the dimensions of real objects, and are presented as a 3D visualization; see Figure 3.

*4.2. Agent*

Agents are objects that act relative to the control system, such as pedestrians and vehicles. The agent is determined to be a vehicle, where a Tesla Model 3 is represented instead of an autonomous vehicle, as shown in Figure 3b, a dynamical model of which is included in CARLA. The agent consists of the sensors and actuators in the vehicle. The sensors in the program are of several types, such as GPS, speedometer, IMU, LiDAR, camera, and so on. To the output of the sensor, noise can be applied, in order to simulate actual noise such as fog, rain, and smoke. The actuators describe such components as the steering wheel, brake, and traction motor.

In this study, LiDAR is included with conventional sensors and is used to observe obstacles and the environment. The attributes were set according to the sensor parameters of the Velodyne HDL-32E [19,22], which obtains 32 channels, and the sensor was mounted in front of the vehicle. The range of detection was set at 40 m. Vehicle motion was performed by the steering wheel, acceleration pedal, and brake pedal. The steering wheel controls the steering angle, which was between ±70 degrees. The acceleration and brake pedals can each be assigned from 0 to 100%.

*4.3. Control System*

The control system can be used for feedback control or otherwise. This study is designed based on the use of a DQN algorithm, which is a kind of reinforcement learning method. The overall system includes path planning, obstacle detection, and the DQN algorithm. The path planning is determined on the straight road, while the DQN algorithm receives sensor data and determines the action to take to control the actuators.

## 5. Deep Q-Network Control

Deep Q-network is a kind of reinforcement learning approach, which was applied to control the agent vehicle, in order to assign actions to actuators such as the acceleration pedal, brake pedal, and steering wheel. This required definitions of the DQN model, its training scheme, and the reward function. Deep Q-learning can work well with complex decisions and requires a huge volume of data, such as large state input, elaborate environment, and delicate action data.

Figure 5 presents the process of Deep Q-learning, which uses a neural network to approximate the Q-value function. The state is given as the input, and possible actions are generated as the output. The DQN model computes the reward function to choose the action for a given state, and store its experiences in a memory buffer. Then, the model uses the experiences from the memory buffer to train the target model, which calculates the best way to maximize its reward.

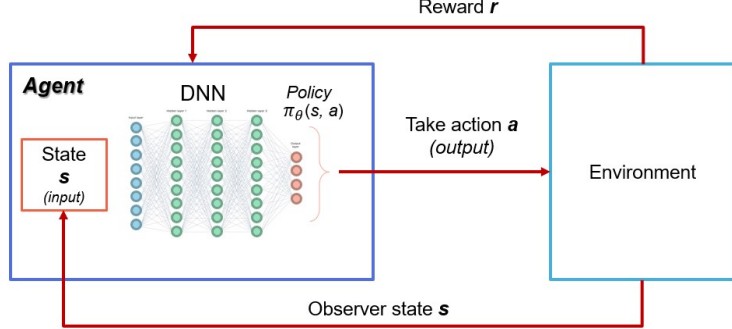

**Figure 5.** Deep Q-Learning process.

To train the DQN algorithm, the best reward and state are collected in the memory buffer, as shown in Figure 6. The remainder of this section is separated into four parts: Model setting, action setting, reward setting, and hyperparameter setting. These four parts

are very important in enabling the DQN model to learn correctly and appropriately for the experiments. The details of designing the control system are provided below.

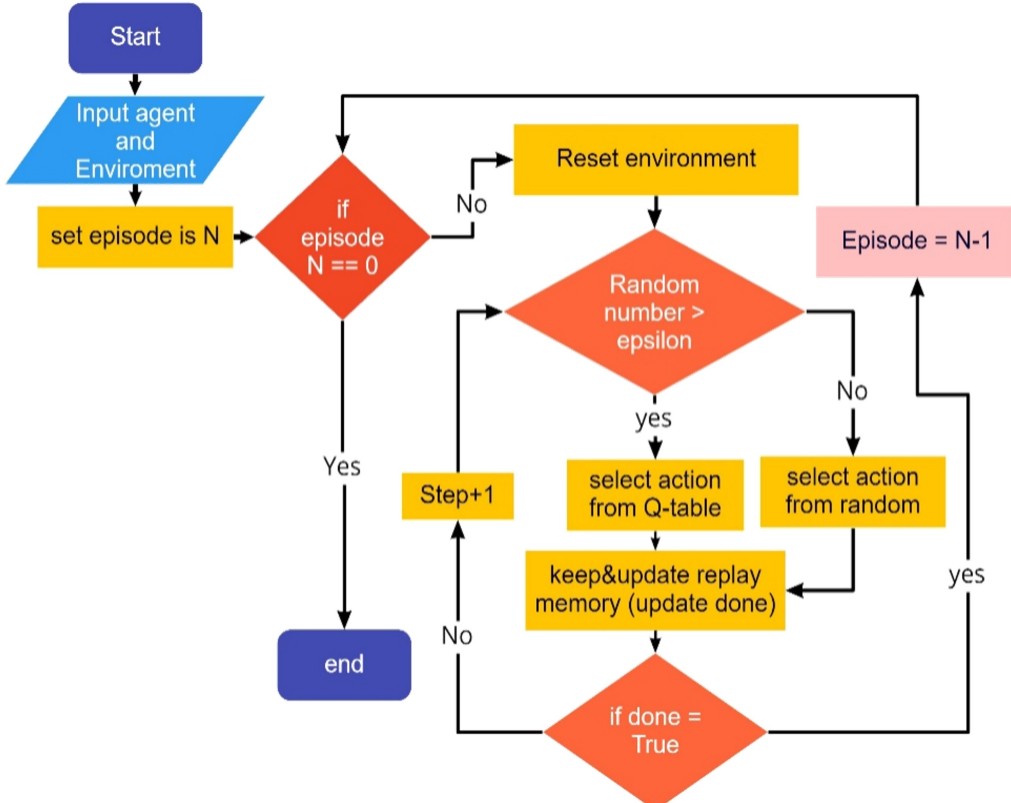

**Figure 6.** Process to select the action in DQN model.

### 5.1. Model Setting

The DQN model consists of dense and hidden layers, which include nodes in the neural network form. Each layer uses an activation function, which matches the type of data that has been designed by the user. If the user does not configure it, the activation function will be designated as a linear function by default. The activation function setting must be considered, in order to minimize the problems of gradient vanishing and explosion.

A total of 32 dense layers are used in the network in this project, with 32 nodes per dense layer, which are re-trained in all the algorithms. The ReLU activation function was applied in the hidden layer; see the settings given in Table 1. Information such as distance from start to destination, the real-time position for calculation in the X-Y coordinates, velocity in the X-Y coordinates, and distance from the vehicle to the obstacle (which comes from the LiDAR sensor) in X-Y coordinates are taken as input data to the DQN algorithm. The LiDAR point cloud, which scopes the areas where the vehicle may potentially collide, is used to find the nearest point, which is later used for calculating the penalty reward.

**Table 1.** List of hyperparameters and their values.

| Parameters | Values | Descriptions |
|---|---|---|
| Number of hidden layers and dense | $32 \times 32 \times 32$ | The number of nodes in the neuron network. In this case, there are 3 hidden layers and 32 dense layers. |
| Activation function | ReLU | Function which defines how the weighted sum of the input is transformed into an output from a node (or nodes) in a layer of the network. |

**Table 1.** *Cont.*

| Parameters | Values | Descriptions |
|---|---|---|
| Replay memory size | 1,000,000 | SGD updates are sampled from this number of most recent steps. |
| Target network update frequency | 400 | The frequency (measured in the number of parameter updates) with which the target network is updated. |
| Discount factor | 0.95 | Discount factor gamma used in the Q-learning update. |
| Learning rate | 0.001 | Tuning parameter that determines the step size at each iteration while moving toward a minimum of the loss function. |
| Initial exploration | 1 | Value of $\epsilon$ in $\epsilon$-greedy exploration. |
| Final exploration | 0.0001 | Final value of $\epsilon$ in $\epsilon$-greedy exploration |
| Final exploration step | 1840 episodes (600,000 steps) | The number of steps over which the initial value of $\epsilon$ is linearly annealed to its final value. |
| Replay start size | 4000 | A uniform random policy is run for this number of steps before learning starts and the resulting experience is used to populate the replay memory. |
| Input data size | 6 | The number of the data that given as input in Q-network |
| Output data size | 4 | The number of the action that given as output in Q-network |

## 5.2. Action Setting

In the action setting stage, we define four actions: straight forward, brake, and steering wheel movement, bounded at ±35 degrees. The acceleration and brake pedals are acted upon relative to the DQN algorithm considerations.

## 5.3. Reward Setting

The reward setting consists of three rewards: the main reward ($R_m$), penalty reward ($R_p$), and extra reward ($R_{ex}$). These should be set as continuous functions, in order to allow the algorithm to learn in the correct and smooth direction. The hyperbolic function was used, with different weights for each reward condition. If the agent can provide a good solution, the reward will be doubled. If the agent provides a bad solution, the reward will be negative. The function of each reward condition has an effect during the training period. The total reward, $R_T$, is given as

$$R_T = R_m + R_p + R_{ex}. \tag{1}$$

The main reward ($R_m$) is designed relative to the distance from the vehicle to the destination ($d_m$), as detailed in Equation (2). This reward is designed based on the quadratic polynomial, determining the farthest position with a negative value and the nearest position having the highest value. Thus, the agent is driven to arrive at the destination. The coefficients for the main reward were determined using the trial and error method, with the maximum reward as 3000 and the minimum being −9500, based on the quadratic polynomial equation.

$$R_m = -a(d_m - b)^2 + c, \tag{2}$$

$$d_m = \sqrt{(x_d - x_v)^2 + (y_d - y_v)^2}, \tag{3}$$

where $d_m$ denotes the distance from the vehicle to the destination; $x_d$ and $y_d$ denote the desired position's $x$ and $y$ coordinates, respectively; $x_v$ and $y_v$ denote the vehicle's, respectively; and the coefficients of $a$, $b$, and $c$ are −2.00, −98.00, and 3000, respectively.

The penalty reward ($R_p$) is computed according to the distance obtained from the LiDAR, as shown in (4). It depends on the minimum distance between the vehicle and obstacles, where obstacles can be detected within the area at a distance of $\pm 3$ m on the lateral axis, distance from the road to 1.3 m on the vertical axis, and distance of 0–70 m on the longitudinal axis. Thus, decreasing the size of the LiDAR point cloud information will affect the process of calculation, allowing the agent to make faster decisions while still effectively detecting obstacles. A penalty reward was created to restrain the motion to the destination while avoiding obstacles. The equation is designed as a fourth-degree polynomial equation, which imitates human behavior. If the LiDAR detects the obstacle at the farthest point, the value of the penalty reward is slightly positive. On the contrary, when the distance between the vehicle and an obstacle is lower than 50 m, the function takes a negative value in the penalty reward, which decreases rapidly when it is lower than 15 m and approaches 0 m (which is the critical point); this affects the algorithm to change its decisions when compared with the main reward. The critical areas are calculated according to the distance required to come to a stop when at a speed of 40 km/h. The coefficients of the penalty reward were determined through the trial and error method and were designed for balancing the main reward and the distance between the vehicle and the obstacle.

$$R_p = a_1 d_l^4 + a_2 d_l^3 + a_3 d_l^2 + a_4 d_l + a_5, \tag{4}$$

$$d_l = \min \sqrt{x_o^2 + y_o^2 + z_o^2}, \tag{5}$$

where, $x_o$, $y_o$, and $z_o$ denote the nearest detected positions, which are scalars of distance vectors from vehicle to obstacle. The coefficients of $a_1$, $a_2$, $a_3$, $a_4$, and $a_5$ are $-0.0011$, 0.15, $-6.60$, 145, and $-3040$, respectively.

The extra reward ($R_{ex}$) is a special reward, which is added to the total reward when the agent can avoid collisions within the episode. This reward is added when the vehicle can brake within 10 m of the obstacle. The design of this reward was aimed at driver behavior. The value of this reward is effective in terms of affecting the choice when making decisions. The reward is given as

$$R_{ex} = b_1 x_l + b_2, \tag{6}$$

where $x_l$ denotes the distance from the vehicle to an obstacle. The coefficients $b_1$ and $b_2$ take values of $-200.00$ and 2200, respectively.

### 5.4. Hyperparameter Setting

The hyperparameter settings can be separated into two parts, relative to the network structure and training algorithm. They are defined in Table 1, where the hyperparameters for the network structure include the number of hidden layers and units, dropout, network weight initialization, and activation function. The hyperparameters for training algorithms include the learning rate, number of epochs, and batch size. $\epsilon$-greedy exploration is an exploration strategy in reinforcement learning that takes an exploratory action with probability and a greedy action [11]. We used an initial exploration value of 1 and final exploration value of 0.0001, with the epsilon value decaying by a factor of 0.995 in every episode. For every 2000 episodes, the DQN model that obtained the highest reward was selected and re-trained with a new initial exploration value (i.e., 0.7), in order to protect against convergence resulting in an incomplete collection of values.

### 6. Case Scenario

One of the most common accident cases in Thailand is where the agent vehicle drives on the main road while an obstacle vehicle drives across the main road, as shown in Figure 7.

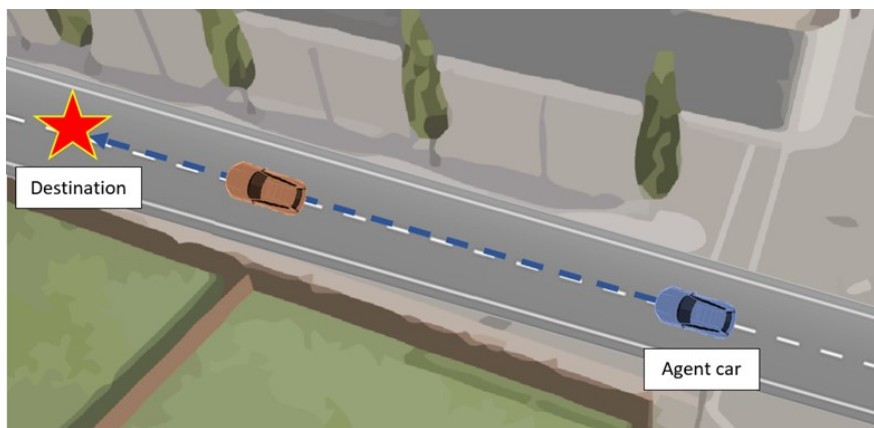

**Figure 7.** The scenario obstacle avoidance in low-speed control.

The agent vehicle cannot brake or avoid in time, making it challenging to design a function for avoiding the obstacle vehicle using the DQN algorithm.

At present, the laws in Thailand on autonomous vehicles do not allow their use in public areas, but they can be applied in private areas with a speed limit of 40 km/h. Therefore, in the case scenario, we considered the initial velocity of the agent car to be 40 km/h and the destination to be 100 m. The front wheel's angle is bounded at ±35 degrees. Then, the agent vehicle is driven to the destination while the obstacle is placed on its trajectory. The agent vehicle should learn to brake or avoid crashing.

## 7. Simulation Results

According to the case scenario, the parameters were configured as shown in Table 1. The DQN model was developed relative to the number of training episodes. The motion was presented by 3D visualization in the CARLA simulator, as shown in Figure 8. The results are shown in Figures 9 and 10, which present the distance from the vehicle to the destination, its velocity, and the reward functions.

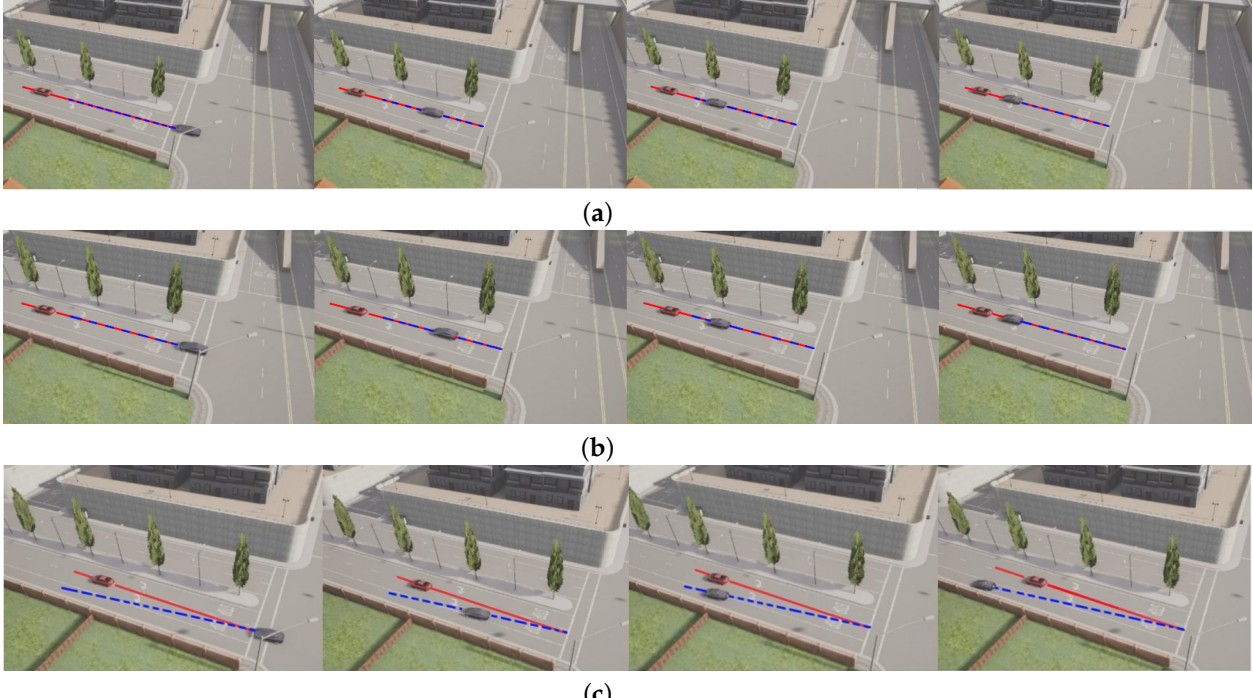

(a)

(b)

(c)

**Figure 8.** *Cont.*

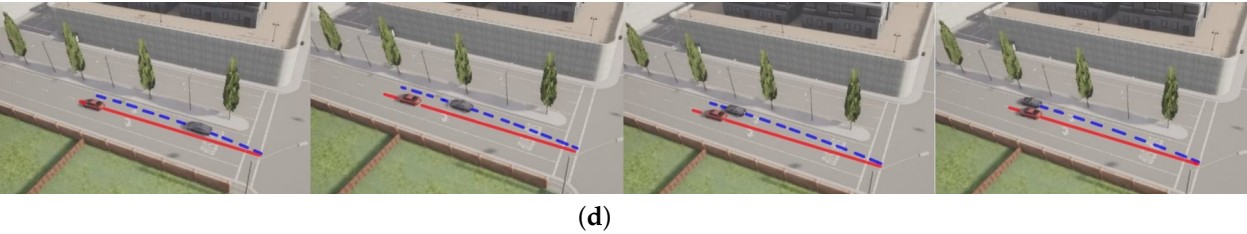

(**d**)

**Figure 8.** The agent performs using DQN model at 1400, 5000, 7000, and 8000 training episodes. (**a**) 1400 training episodes. (**b**) 5000 training episodes. (**c**) 7000 training episodes. (**d**) 8000 training episodes.

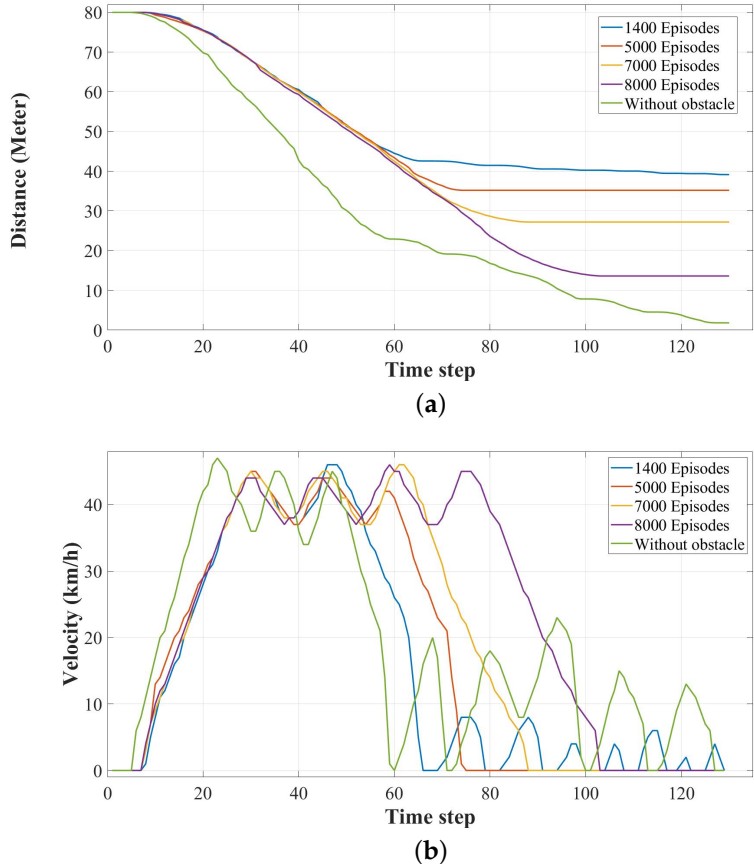

(**a**)

(**b**)

**Figure 9.** The motion of the agent relative to the number of training episode for straight motion test. (**a**) Distance from agent to destination. (**b**) Velocity of agent.

Considering the distance and velocity of the agent vehicle in Figure 9, the distance from the vehicle to the destination is provided in Figure 9a. As the number of training episodes increased, the final position of the vehicle came closer to the destination. The decision of the control system may switch between stopping and moving forward. Motion after stopping is shown by the blue line in Figure 9, which presents the vehicle motion according to the DQN model at 1400 training episodes. In the other cases, the vehicle permanently stopped. Therefore, the decision of the control system using 1400 training episodes may not have been sufficient for obstacle avoidance, as some nodes in the DQN model may have still had blank nodes. Thus, the control system may continue to attempt to find a better path, causing it to move forward after stopping.

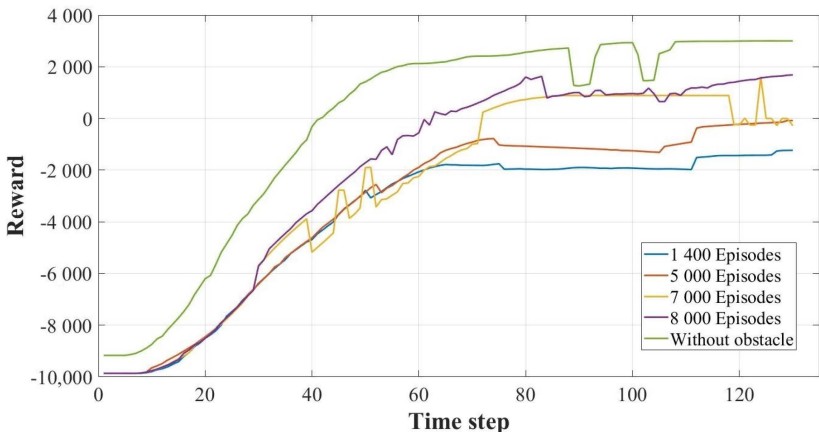

**Figure 10.** Reward for each number of training episode for straight motion test.

The velocity profile in each case is shown in Figure 9b. The initial velocity in all cases was 40 km/h, and the velocity using the DQN models at 1400 and 5000 training episodes decreased with low deceleration, after which the deceleration increased to zero velocity. The velocity using the DQN models at 7000 and 8000 training episodes decreased with almost constant deceleration to zero velocity.

Figure 10 illustrates the reward during the motion from the initial position to the goal. The rewards increased when the vehicle came closer to the destination without any collision. For this reason, the rewards for the DQN models at 1400, 5000, and 8000 training episodes increased to convergent values, and the oscillation of rewards may have occurred due to the penalty reward, as the minimum distance from the vehicle to an obstacle may be affected by the shape of the obstacle.

As mentioned above, the number of training episodes affected the vehicle's behavior. The control system used DQN models trained to 1400, 5000, 7000, and 8000 training episodes, respectively. The demonstration is shown in Figure 8. The vehicle control systems using DQN models at 1400 and 5000 training episodes could stop before colliding with the obstacle, but this was not good enough to avoid the obstacle, as shown in Figure 8a,b, respectively.

With DQN models trained at 7000 and 8000 training episodes, the vehicle behavior was improved, such that the vehicle could avoid the obstacle, as shown in Figure 8c,d. The vehicles avoided the obstacle by decelerating and could stop at the destination using the steering angle for control. The vehicle using the model at 7000 training episodes immediately avoided the obstacle after it was detected by LiDAR. The vehicle using the model at 8000 training episodes avoided the obstacle when it neared the obstacle, rather than in other cases.

According to the reward graph, as shown in Figure 10, the values for training without obstacles between the time steps 85–95 and 100–110 represented the attraction for DQN-based decision making, with balancing of the coefficients for the main reward and penalty reward. Thus, even if the LiDAR sensor detected an obstacle in the lane, it did not have an effect on the decision as long as it was not in the critical area (i.e., within 15 m). Comparing training with obstacles, the reward result also revealed the trend that the results were better when the number of training episodes was increased. When comparing the reward graph between 7000 and 8000 episodes of training, the shape of the former fluctuated more than with 8000 episodes of training, due to disturbance by the LiDAR sensor. Therefore, the DQN algorithm proposed in this study attempted to avoid sudden changes in reward value.

In [21], an autonomous vehicle control system was tested on the CARLA simulator by evaluating three methods: Modular pipeline (MP), imitation learning (IL), and reinforcement learning (RL). The performance of the reinforcement learning method was not satisfactory. Due to the driving scenarios, the agent must deal with vehicle dynamics and

complicated perceptions in a dynamic environment during vehicle movement. Therefore, the DQN algorithm was applied in our driving case scenario.

The limitation of this work was the use of a specific case scenario, involving avoiding a static obstacle in front of the vehicle. If the obstacle is a dynamic obstacle or obstacles are on the side of the vehicle, the proposed algorithm may not be successful. Different boundary conditions must be set in a new critical data set, which includes collision risks, and the algorithm should be re-trained.

## 8. Conclusions

According to our results, we successfully developed a control system using the DQN algorithm for obstacle avoidance. CARLA was used to present the results through visualization and graphs. The vehicle motion using the DQN model at a higher number of training episodes was better than at a low number of training episodes.

To train the DQN model, the best training result was recognized as being related to the replay memory size. If the model is trained by TensorFlow version 1, the recognized memory may be divergent and some memories may not present the best results, thus causing the failure of the simulation results. Therefore, TensorFlow version 2.8 should be used, instead of version 1, as the results are better than those obtained with the first version.

It should be noted that $\epsilon$-greedy exploration influences the quality of replay memory. If a small initial exploration value is used, the model may not find a new solution. If the final exploration value is a large value, the model may not make the decision by itself. For the 0.5 $\epsilon$-greedy exploration value, the model slowly improves, but it may not be suitable, according to the scenario case, and takes a lot of time in training. Thus, the value in this project was designed to slowly decay from 1 to 0.0001, as detailed in Table 1. The purpose of this was to allow the model to search for a new solution and then, after that, to gradually increase in order to make the decision by itself. The replay memory size was designed to make sure that it can collect all the states sufficient for generation in the DQN model.

This work addresses just one of the functions of an autonomous control system. Therefore, we intend to apply DQN models to the other functions and combine them for fully autonomous vehicle control in future work.

**Author Contributions:** Conceptualization, J.R., D.P. and P.K.; methodology, J.R., W.T., P.K. and D.P.; software, W.T. and P.K.; validation, J.R., P.K., D.P. and W.T.; formal analysis, D.P.; investigation, J.R.; resources, J.R.; data curation, W.T.; writing—original draft preparation, W.T.; writing—review and editing, W.T., D.P. and J.R.; visualization, W.T.; supervision, J.R. and D.P.; project administration, W.T.; funding acquisition, J.R. and D.P. All authors have read and agreed to the published version of the manuscript.

**Funding:** This research project is supported by Thailand Science Research and Innovation (TSRI) Basic Research Fund: Fiscal year 2022 under project number FRB650048/0164.

**Data Availability Statement:** Not applicable

**Conflicts of Interest:** The authors declare no conflict of interest.

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
