# Peer review of "Design of Obstacle Avoidance for Autonomous Vehicle Using Deep Q-Network and CARLA Simulator"

_wevj, doi:10.3390/wevj13120239_

Round 1

Reviewer 1 Report

This paper presents a reinforcement learning-based obstacle avoidance method for autonomous vehicles. The comments are listed as follows:

 1 There are some grammar errors, such as in Line 112-113, 140. Please carefully check and correct them.

2 Please specify the exact input for the DQN algorithm. Image? Lidar point cloud? The abstractive message such as velocity, position? Please clarify them.

3 In Fig. 6, the author used the epsilon-greedy strategy to balance exploration and exploitation. The author used a random number. However, it should be a sample of the normal distribution to decide whether to investigate or not. Please elaborate.

4 Please give more reasons for the reward shaping process, such as (4). What are the criteria for parameter selection and calibration?

5 As the steering angle is one of the outputs from the DQN, I suggest the author show the exact vehicle trajectory when it performs obstacle avoidance. Now, it is missing in the current text.

Reviewer 2 Report

The paper presented an experiment and its result for autonomous vehicles avoiding on-road obstacles with deep-Q learning. The work is based on simulations and some of the results are presented.

However, there a huge room for improving the quality of the paper:

1. There are plenty of English grammatical error throughout the manuscript. This reduces the readability of the paper dramatically. Please see the attached separate comment document for a few examples I have highlighted in the abstract and the introduction section. The authors are recommended to use some professional services for drafting the paper and do a through proof-reading before any attempt of resubmission.

2. The Abstract needs to be rewritten to make it like an abstract. It should be a brief and concise high-level description of your work, the advantages and disadvantages of your methods, and the achievements. As it stands now, it is more like an summary of everything the authors have done for the experiment.

3. The paper needs to have a section of "Related work" in this area. There are many research work in using deep learning for obstacle/collision-avoidance as well as safety driving policy using multiple sensors and neural networks. The authors should understand what have been done and what have been achieved in this hugely important area before they embarking on the design of their methodology and the description of their own work.

4. Section 3 should be shortened in introducing the different technologies, which is not the key points of discussion in this paper. Readers want to know what the development in this area and how your work is related and compared with others.

5. The authors should be clear in section 4, about the deep-Q network: whether the network architecture is designed by themselves or they just use a typical DQ structure for their experiment.

6. The simulation results presented in Section 6 need more explanation and discussion. How do your results compare with others? Is there anything you can do to make the result better or the performance of your algorithm improved, for example, using less time/computing resources/short distance to stop/avoid the obstacle? Right now, it presented in a way that this is it and that's it. Authors should think and ask "why", "what" and "how".

7. What are the limitations of the methods, experiment and results in the presented work?

8. Please see the attached comment file for more details.

Reviewer 3 Report

This paper reports the results of obstacle avoidance using DQN. It is well-written and easy to follow. But it ha=s some weaknesses.

1) The contribution and novelty are not very significant. It is not the first time to use DQN for obstacle avoidance. 

2) It's better for authors to explain why DQN is used. Because many excellent reinforcement learning methods have been proposed recently, such as A3C, double DQN and so on. DQN is classic.

3)It's better for authors to use variables to denote the numbers in equations 2,4 and 6. And explain why the variables are set to these numbers.

4) To demonstrate the superior performance, comparing your methods with existing methods is necessary.

5) Both quantitative and qualitative results are needed. But only qualitative results are given.

Round 2

Reviewer 1 Report

All concerns have been well justified in the revised version. 

Author Response

The new version has revised some grammatical errors and additional detail in section7.

Reviewer 2 Report

The quality of the manuscript v2 is improved in comparison with v1. However, the manuscript needs further revision before it can be accepted for publication. The general comments are as follows:

1. There are too many English grammatical errors, wrong word choices and awkward sentences. Extensive revision in English is needed. 

2. The authors needs to describe or explain why certain parameter values are selected or used in the experiment, such as the values of a, b, and c in (2), in line 196. 

3. The trajectory of agent vehicle presented in Figure 8c and 8d contradicts with the author's own declaration in line 180-181, where the author defined only two actions of straight forward and brake. However, the Figure 8c/d showed the agent vehicle can sway away from its direction without colliding with the obstacle. The author needs to explain why and how this could happen.

4. The paper does not include any kind of comparison of results with other research in this area. It should include some comparison in the section 7 of simulation result. Is it any better than other algorithms on the same simulation platform? 

5. There are more comments and change suggestions in the attached summary of comments file. 

Round 3

Reviewer 2 Report

The paper is now in good condition for publication.

Author Response

manuscript was already revised